# Subjects are not all alike: Eye-tracking the agent preference in Spanish

**Beatriz Gómez-Vidal** [1]*, **Miren Arantzeta**[1], **Jon Paul Laka**[2], **Itziar Laka**[1]

**1** The Bilingual Mind Research Group, Department of Linguistics and Basque Studies, University of the Basque Country (UPV/EHU), Vitoria-Gasteiz, Basque Country, Spain, **2** University of Deusto (DBS), Bilbao, Basque Country, Spain

* beatriz.gomezv@ehu.eus

**Data Availability Statement:** We have uploaded the minimal anonymized datasets necessary to replicate our findings to a stable and public repository. These datasets can be found under the same link that was provided in earlier stages of

## Abstract

Experimental research on argument structure has reported mixed results regarding the processing of unaccusative and unergative predicates. Using eye tracking in the visual world paradigm, this study seeks to fill a gap in the literature by presenting new evidence of the processing distinction between agent and theme subjects. We considered two hypotheses. First, the Unaccusative Hypothesis states that unaccusative (theme) subjects involve a more complex syntactic representation than unergative (agent) subjects. It predicts a delayed reactivation of unaccusative subjects compared to unergatives after the presentation of the verb. Second, the Agent First Hypothesis states that the first ambiguous NP of a sentence will preferably be interpreted as an agent due to an attentional preference to agents over themes. It predicts a larger reactivation of agent subjects than themes. We monitored the time course of gaze fixations of 44 native speakers across a visual display while processing sentences with unaccusative, unergative and transitive verbs. One of the pictures in the visual display was semantically related to the sentential subject. We analyzed fixation patterns in three different time frames: the verb frame, the post-verb frame, and the global post-verbal frame. Results indicated that sentential subjects across the three conditions were significantly activated when participants heard the verb; this is compatible with observing a post-verbal reactivation effect. Time course and magnitude of the gaze-fixation patterns are fully compatible with the predictions made by the Agent First Hypothesis. Thus, we report new evidence for (a) a processing distinction between unaccusative and unergative predicates in sentence comprehension, and (b) an attentional preference towards agents over themes, reflected by a larger reactivation effect in agent subjects.

## 1. Introduction

### 1.1 Unaccusativity and thematic roles

The split of intransitive predicates into unaccusatives (1) and unergatives (2) is the object of a long-lasting discussion in theoretical linguistics [1–4]. Theoretical approaches concur that unaccusative predicates take a *theme* as a single argument (1), whereas unergatives take an *agent* (2).

submission and revision of our manuscript: <https://osf.io/s4yrx/>. No other step is necessary to access or download the datasets.

**Funding:** This work was supported by the Spanish Ministry of Science, Innovation and Universities (BGV, FPU18/04268; BGV, MA & IL, PID2019-104016GB-I00); the University of the Basque Country (UPV/EHU) (MA, ESPDOC18/74); the Basque Department of Education, Universities and Research (BGV, MA & IL, IT1439-22); and the BBVA Foundation for Researchers and Cultural Creators (MA, 2021). The funders had no role in study design, data collection and analysis, decision to publish, or preparation of the manuscript.

**Competing interests:** The authors have declared that no competing interests exist.

1. [$_S$ [$_{NP}$ The girl]$_i$ [$_{VP}$ fell $t_i$.]]

2. [$_S$ [$_{NP}$ The girl] [$_{VP}$ ran.]]

This difference has important implications for syntactic structure. Formal approaches [1–4] characterize the syntax of unaccusative predicates (1) as more complex than that of unergatives (2). More specifically, the Unaccusative Hypothesis [1] (henceforth UH) claims that unaccusative subjects start the derivation as objects, and then move to subject position. By contrast, unergative arguments are generated in subject position. This additional dependency in the structure of unaccusative predicates (1) compared to unergatives (2) is shown above in terms of syntactic movement within a Government and Binding framework [2, 5]; note that the unaccusative argument leaves a trace at its base position.

Research in Psycholinguistics testing the UH has found processing differences between unaccusative and unergative predicates. Studies conducted in a variety of methods and languages show evidence of a slower or less accurate processing of unaccusative sentences compared to unergatives [6–14]. They report several phenomena associated with the processing of unaccusative sentences as compared to unergatives: (a) larger reaction times for unaccusatives in cross-modal lexical priming studies [7, 10]; (b) higher error rates for unaccusatives in populations with and without aphasia [9, 11, 12]; and (c) increased brain cortical activation for unaccusatives in neuroimaging studies [13, 15]. Studies using cross-modal lexical priming techniques [7, 10, 15] specifically explored whether unaccusative and unergative subjects were reactivated after the verb. Results showed a late post-verbal reactivation of the subject (around 750 ms after verb offset) in unaccusative predicates, but not in unergatives. According to the Trace Facilitation Hypothesis [7], speakers reactivate the mental representation of the referent at the point where its trace is encountered [16–18]. For this reason, their results [7, 10, 15] were interpreted as evidence that the unaccusative argument had deposited a trace within the VP due to syntactic movement, whereas the unergatives argument had not, as predicted by the UH.

However, more recent research using continuous measures of language processing indicates that post-verbal reactivation is not exclusive to unaccusative subjects, since the unergative subjects also undergo reactivation after the verb, with an earlier reactivation than for unaccusatives [8, 19]. In a seminal study using eye tracking in the Visual World Paradigm (henceforth VWP), Koring et al. [19] found two different patterns of subject reactivation in Dutch sentences with preverbal subjects: an early reactivation of the unergative subject (peaking around 300 ms after verb offset), and a late reactivation of the unaccusative subject (peaking around 950 ms after verb offset). As a consequence, these authors [19] claim that post-verbal reactivation of a preverbal argument occurs independently of its thematic role, because argument reactivation is needed for the integration of an argument with its verb into a single semantic representation. Their results [19] were interpreted as evidence for the UH, since a late peak in the reactivation of unaccusative subjects is compatible with added steps in the syntactic interpretation of unaccusative subjects compared to unergatives.

Many studies [7, 8, 10, 15, 19] have interpreted the general finding of observing a processing distinction between unaccusative and unergative predicates as evidence for the UH. However, we claim that an alternative interpretation is possible attending to a generalized processing distinction between agents and themes. More specifically, the Agent First Hypothesis [20–22] (henceforth AFH) proposes a preference for agents over themes; this is compatible with finding processing differences between unaccusative and unergative predicates. In fact, there is a significant body of research in psycholinguistics and cognitive science to support such a distinction. Agent and theme roles correspond with two abstract and relatively salient categories that are accessed and processed rapidly and robustly in processing [23–33]. This general

finding has been reported in studies investigating adult and child populations, as well as during exposure to visual and/or linguistic events. More importantly, there is evidence that agents and themes follow distinct processing patterns. For example, agent subjects are preferred as the first argument in ambiguous syntactic contexts across a variety of languages [20, 22, 34–36]. Agents are also recognized faster than themes in visual events both by neurotypical populations [37–40] and by people with agrammatic aphasia [41]. In acquisition literature, studies show that children learn to detect the agent in a given event before they learn to detect the theme [26, 27, 42–44]. Overall, this large body of evidence [40, 45] points towards agents being more distinct and cognitively more salient than themes; this is one of the crucial claims of the AFH.

A recent study in English by Huang and Snedeker [46] conducted a close replica of Koring et al. [19] in three different experiments, failing to replicate any aspect of Koring et al.'s [19] findings. These authors suggest that Koring et al.'s [19] results are unreliable due to the method used for data analysis (i.e., Growth Curve Analysis), which they argue to be highly anti-conservative and ill-suited for psycholinguistic studies. Here, we aim to contribute to this debate on argument structure and thematic role processing by presenting our results from a Spanish close replica of Koring et al. [19], which also bears close resemblance to the study recently conducted by Huang and Snedeker [46]. We consider two hypotheses, the UH and AFH, as possible accounts for findings reported in previous literature which observe a processing difference between unaccusative and unergative predicates. Our aim is to discriminate between the two, since they make different predictions for our data. We investigated the processing of agent and theme subjects in unaccusative, unergative and transitive sentences in Spanish using eye tracking in the VWP. This methodology [47–49] consists in the simultaneous presentation of auditory and visual stimuli. The basic assumption behind the VWP is that eye fixations on visual targets are automatically guided by referentially-related linguistic stimuli. Hence, the likelihood of fixating on specific visual objects increases attending to the semantic and phonological relation between the visual object and the auditory linguistic message [50–52]. This methodology is a reliable measure of how different elements in the linguistic stimuli are activated (and reactivated) in participants' mental representation. As such, it is able to reveal how changes in attention to linguistic elements unfold during linguistic processing.

We presented 44 Spanish native speakers with spoken sentences while they viewed a display of pictures on a screen. In test trials, the visual target (e.g., *cheese*) was semantically related to the subject in the spoken sentence (e.g., *ratón* 'mouse'). We measured the proportion of fixations toward the visual target during the presentation of the spoken sentence to investigate argument reactivation after the verb. Argument reactivation is a processing phenomenon by which preverbal arguments become activated again in the mental representation of listeners once the sentential verb is encountered [7, 8, 10, 15]. It is argued that preverbal subjects must be reactivated after the presentation of the verb to integrate the argument and the verb into a single mental representation [19]. We measured the proportion of looks to a semantically-related visual target to detect the magnitude and time course of the reactivation of the sentential subject after the presentation of the verb.

## 1.2 Hypotheses

We consider two hypotheses in our experiment. First, the UH [1] claims that unaccusative subjects have longer syntactic derivations than unergative subjects. According to this hypothesis, unaccusative subjects are born as objects of the verb and must be promoted to subject function; Burzio [2] reinterprets this as syntactic movement, where the initial object moves to subject position leaving a trace in its original position. By contrast, unergative subjects are born in

subject position and do not require additional interpretive steps like identifying the gap and assigning an argument to it. By extension of the UH [1], and in line with previous research [7, 8, 10, 15], Koring et al. [19] predict that unaccusative subjects will display a delayed reactivation compared to unergative subjects. Thus, two distinct reactivation patterns should be observed: an early reactivation for unergative subjects, and a late reactivation for unaccusative subjects. This predicts a difference in the time course of the reactivation effect between conditions, but no differences in the magnitude of the effect. Previous offline research [7, 8, 10] have situated the unaccusative subject reactivation effect at around 750 ms after verb offset. Because the human eye needs approximately 200 ms to program and initiate movement in reaction to auditory stimuli [53], and based on the predictions of the UH, we should find a peak of gaze fixations signaling a reactivation of the unaccusative subject around 950 ms after verb offset, coinciding with Koring et al. [19].

Second, we consider the Agent First Hypothesis [20–22] (henceforth AFH). This hypothesis is strongly related to similar proposals in psycholinguistics, such as the Subject First Hypothesis [54–56], the NVN strategy and the Actor Strategy [57, 58]. The NVN strategy assumes a mapping of semantic roles to syntactic positions following an agent-action-patient order. The Actor Strategy claims that the language processing system prioritizes identifying the agent in the event. The Subject First Hypothesis claims that the first ambiguous NP in a given sentence is preferably processed as a subject, and has been supported by evidence from a number of studies [35, 36, 59, 60]. However, and because this hypothesis fails to take into account the thematic role of the subject, most of this body of evidence is also compatible with the AFH. The AFH claims that the first ambiguous NP in a given sentence is preferably processed as an agent, a claim that has been supported by some experimental research [20, 22]. In our study, and based on the array of evidence pointing toward the salience of agents compared to themes from a cognitive point of view, we assume the following formulation of the AFH: that the first ambiguous NP in a given sentence is preferably processed as an agent, because listeners display an attentional preference toward agents as compared to themes. In our data, the AFH predicts that agent subjects (unergative and transitive conditions) will display a larger reactivation effect than theme subjects (unaccusative condition) after the verb. This is so because, in the VWP, fixations on a visual object reveal how a related entity is being activated, processed, or otherwise being attended to in the participant's mental state. We argue that observing a difference in the magnitude of the reactivation effect (i.e., a significantly higher proportion of looks to the visual target in one condition compared to the other) can be interpreted as a difference in the attentional resources that listeners are devoting toward the reactivated element in their mental state.

## 2. The present study

Using an experimental design closely following that of Koring et al. [19], we explored the processing patterns of unaccusative, unergative and transitive subjects in Spanish to test the predictions of the UH [1] and the AFH [20, 22]. To achieve this, we created SV(O) sentences where we measured the time course of subject reactivation upon encountering the verb by means of eye tracking in the VWP. The types of predicates explored differed slightly with respect to the original study: while Koring et al. [19] investigated sentences with unaccusative, unergative and mixed verbs, we decided to incorporate sentences with unaccusative, unergative and transitive verbs into our study. The reason for this change has to do with the hypotheses that we contemplated. Because the AFH makes its predictions based on the thematic role of the argument (in this case, the sentential subject), it predicts no differences in the reactivation pattern of unergative and transitive subjects, which are both agents.

## 2.1 Methodology

We presented participants with auditory sentences and visual input simultaneously. To measure subject reactivation in our three predicate conditions, we created sentences with preverbal subjects in which we expected to find subject reactivation after the verb. We were interested in measuring subject reactivation, since it indicates how arguments are being processed regarding the argument structure that they participate in. We measured argument reactivation after verb onset by monitoring participants' eye fixations to a visual target that was semantically related to the sentential subject in the spoken sentence (e.g., sentential subject *mouse*, related to a depiction of a cheese).

## 2.2 Stimuli

The present study had a 3 x 2 design with two independent variables: (i) verb type (unaccusative, unergative, transitive), and (ii) trial type (test, control). In test trials, the sentential subject shared a strong semantic relation with the target drawing (e.g., subject *mouse*, related to a depiction of a cheese), thus allowing us to measure subject reactivation. By contrast, in control trials the sentential subject did not share a strong semantic relation with the target drawing (e.g., subject *chimpanzee*, unrelated to a depiction of a cheese), thus serving as a baseline for fixations to the target drawing. The dependent variable was the duration of eye fixations on the visual target. Stimuli comprised 170 trials, consisting of 170 unique spoken sentences paired with four black-and-white line pictures from the International Picture Naming Project website [61]. There were 120 experimental trials in total, including 60 test trials and 60 control trials. The other 50 were filler trials. Experimental trials were distributed evenly across 2 lists of stimuli. Each list of stimuli consisted of 30 test trials, 30 control trials and 50 filler trials. The same set of filler trials was used for both lists.

Linguistic stimuli consisted of spoken sentences recorded by a female native speaker of Spanish in a soundproof booth at a comfortable speaking rate. Experimental sentences had a mean length of 37 syllables (ranging from 31 to 46 syllables), a mean duration of 4244 ms and an average speech rate of 4.8 syllables per second (unaccusative condition: mean length 28 syllables, mean duration 5935 ms, average speech rate 4.6 syllables per second; unergative condition: mean length 29 syllables, mean duration 5829 ms, average speech rate 4.9 syllables per second; transitive condition: mean length 34 syllables, mean duration 6953 ms, average speech rate 4.9 syllables per second). Experimental sentences were structured into 4 (in unaccusative and unergative sentences) or 5 (in transitive sentences) regions of interest or ROIs (see Table 1). ROIs 3 and 4 were the relevant regions for the analysis, since they comprised the presentation of the verb and the post-verbal region. These correspond to the general point in time in which preverbal argument reactivation is to be expected. Following Koring et al. [19], the number of syllables in ROIs 2 and 4 was thoroughly controlled to better align all experimental sentences at verb offset for the analysis. The average number of syllables in ROI 2 was 7.8, ranging from 7 to 9; the average number of syllables for ROI 4 was 13.8, ranging from 13 to 15.

**Table 1. Structure of experimental sentences by ROIs.**

| ROI number | ROI name | Content |
|---|---|---|
| 1 | Introduction | A framing sentence including a variation of "[Someone] said that. . ." |
| 2 | Subject | The sentential subject NP, including a PP or AdjP that modified it. |
| 3 | Verb | The experimental verb (unaccusative, unergative or transitive). |
| 4 | Post-Verb | A post-verbal Adjunct (of manner, time, place. . .). |
| (5) | (Object) | The sentential object NP (only in the case of transitive sentences). |

An example of an experimental transitive sentence is given in (3). Square brackets signal the different ROIs in which experimental sentences were structured (see Table 1 for a complete account).

(3) [La madre dijo que]$_1$ [la peluquera de grandes ojos verdes]$_2$ [contó]$_3$ [cuidadosamente y de manera muy pausada]$_4$ [el número de asistentes.]$_5$

'[The mother said that]$_1$ [the hairdresser with big, green eyes]$_2$ [counted]$_3$ [carefully and in a very slow manner]$_4$ [the number of attendees.]$_5$'

There were two types of experimental trials: test and control, which were created in pairs. Each pair consisted of one test and one control trial. Within each pair, the stimulus was kept identical in both trials, except for one word in the spoken sentence: the subject. In test trials (4), the subject was semantically related to the visual target in the visual display (e.g., subject *mouse*, related to a depiction of a cheese). In corresponding control trials (5), the stimulus was identical except for the subject, which was changed (e.g., subject *chimpanzee*, unrelated to a depiction of a cheese). This was done so that the sentential subject was semantically related to the visual target in test trials, but not in control trials, thus ensuring that control trials would serve as a baseline for fixations to the visual target. An example of a pair of test and control sentences is given in (4) and (5), respectively. Both sentences are paired with the same visual display (Fig 1), and only differ in their sentential subject. Square brackets signal the different ROIs in which experimental sentences were structured (see Table 1 for a complete account).

(4) [La señora dijo que]$^1$ [el ratón negro, peludo y grande]$^2$ [cayó]$^3$ [ese día por las escaleras del edificio.]$^4$

'The lady said that the big, hairy, black mouse fell that day down the stairs of the building.'

(5) [La señora dijo que]$^1$ [el chimpancé negro, peludo y grande]$^2$ [cayó]$^3$ [ese día por las escaleras del edificio.]$^4$

'The lady said that the big, hairy, black chimpanzee fell that day down the stairs of the building.'

The remaining trials were filler trials; in these, one of the words in the spoken sentence was directly matched by one of the drawings in the visual display. Filler trials were highly varied in their word order, syntactic structure and length of constituents. This was done in order to minimize the possibility of participants predicting the moment in which critical words would be presented in test and control trials.

The strength of the semantic relations between sentential subjects and visual targets (e.g., *mouse–cheese*) was determined by means of a norming study conducted online using the Ibex 0.3.8 platform [62]. Fifty-five voluntary Spanish native speakers rated the strength of the semantic relation between two nouns or a noun and a verb on a scale from 0 to 5. We selected strongly-related pairs of nouns (e.g., *mouse–cheese*) for the test sentences (i.e., pairs that received a mean rating of 4 or higher) and weakly-related pairs of nouns (e.g., *chimpanzee–cheese*) for the control sentences (i.e., pairs that received a mean rating of 2 or lower). This allowed us to distinguish eye fixations on the visual target in test trials, which were mediated by the semantic relation between the sentential subject and the visual target, from fixations on the visual target in control trials, which were not mediated by any semantic relationship and were therefore random. We also assessed the strength of the semantic association in pairs of a noun and a verb to ensure that the verbs were not especially related semantically to the sentential subject, which we believed could potentially affect argument reactivation. Thus, all noun and verb pairs selected received a mean rating of 2 or lower. Detailed results of the noun-noun

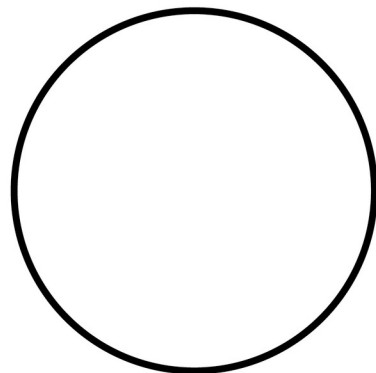
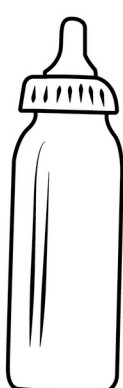
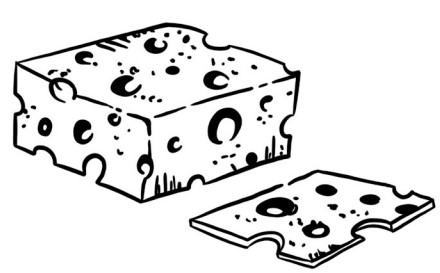
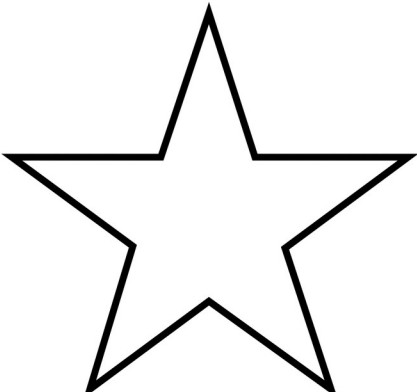

**Fig 1. Visual display paired with test sentence (4) and control sentence (5).** The visual target is cheese, which is related to the sentential subject *ratón* 'mouse' in sentence (4), but unrelated to the sentential subject *chimpancé* 'chimpanzee' in sentence (5).

ratings and noun-verb ratings from the online norming study are provided in the Supporting Information.

## 2.3 Participants and procedure

Forty-four students from the University of the Basque Country (UPV/EHU) participated in the experiment (33 female; mean age 22.6; *SD* 3.6). They were all native speakers of Spanish with normal or corrected to normal vision. This study was approved by the ethics board for human research of the University of the Basque Country (CEISH-UPV/EHU). Prior to their participation, subjects were properly informed of the procedure and indicated their written consent to participate by signing the informed consent paperwork. They were paid 6  for their participation.

The experiment took place in a soundproof booth located at the Micaela Portilla Research Center. Participants were seated on a chair with their eyes about 60 cm from a 24" viewing monitor, set at a resolution of 1024 x 780 pixels. Participants wore binaural headphones to listen to the spoken sentences. The experiment was conducted using Tobii Studio 3.1.0 software

[63], and eye movements were recorded by a Tobii X120 desktop eye tracker sampling at 120 Hz. Each session began with a calibration procedure with nine fixation points. Participants were told that they would hear some recorded sentences through the headphones while static visual displays showed on the screen. They were instructed to listen to the sentences very carefully while looking freely into the monitor. Participants were asked to fixate on a centrally-located cross that appeared between trials. This was done in order to reduce noise in the data, i.e., looks to a particular region of the screen before trial onset. Throughout the experiment, we did not check the status of the eye movements before sentence onset, but this did not pose a problem for our results due to (a) the randomization of the location of the visual target, (b) the use of control trials as a baseline for fixations in test trials. Participants did not have to perform any other task besides listening to the sentences and looking at the pictures. There was no cover story or recurring theme in the experiment. There was 1 second of silent previsualization of the visual stimuli in each trial before the onset of the sentence. Then, the spoken sentence played while the visual display remained on the screen. After the end of the sentence, there were another 2 seconds of silence before the visual display disappeared and the fixation cross appeared for 0.6 seconds on the center of the screen, after which another trial began. Stimuli presentation was randomized per participant. The entire experiment lasted 20 minutes.

## 3. Data

### 3.1 Data processing

We followed the procedure described by Koring et al. [19]. Only experimental (test and control) trials were selected for data analysis. Filler trials were discarded. Data classified as "saccades", "unclassified", and instances of track loss, which represented 22.9% of the data, were eliminated from the dataset. This value indicated that collection of relevant data for our analysis was successful around 80% of the time. Each experimental sentence was time-stamped at the onset of each ROI. Unaccusative and unergative sentences differed from transitive sentences in that their sentence offset took place earlier, given that transitive sentences included an additional ROI (5) for the sentential object NP. However, this difference did not interfere in the analysis because fixations to the visual target were analyzed in comparable time windows across all three conditions (see section 3.2). All trials were centered at verb offset, i.e., around the point in time when argument reactivation was expected to take place. As a result, verb offset corresponded with the 0 ms value in all trials.

To proceed with the data analysis, we first needed to verify that any increase in the fixations to the visual target in test trials was due to the activation of the sentential subject. We calculated the proportion of fixations to the target picture during the auditory presentation of the linguistic stimuli in each ROI. We created a linear mixed model containing three-way interactions for verb type (unaccusative, unergative, transitive), ROI (Introduction, Subject, Verb, Post-Verb, Object (only for the transitive condition)), and trial type (test, control) as fixed effects; stimuli and participant variables were included as random effects. Least-Squares Means (lsmeans) were calculated, and pairwise comparisons were carried out. Effects were considered significant at the $p < .05$ level. Results are shown in Table 2. The analysis was conducted using R Statistic 3.6.2 software [64] using the lme4 package [65] and the lsmeans package [66]. Results showed that in all experimental conditions, participants fixated more on the visual target in test trials than in control trials since the presentation of the Subject ROI onward (i.e., during the presentation of Subject, Verb, Post-Verb, Object), and not before (Introduction) (see Table 2). These results confirmed that gaze fixations to the visual target in test trials were motivated by the semantic relation between the visual target and the sentential

**Table 2. Pairwise comparisons of gaze fixations to the target by ROI and verb type.**

| ROI name | Unaccusative | | | Unergative | | | Transitive | | |
|---|---|---|---|---|---|---|---|---|---|
| | Est. | SE | p | Est. | SE | p | Est. | SE | p |
| Introduction | -0.0265 | 2.94 | 0.9928 | 1.8943 | 2.94 | 0.5188 | 0.493 | 2.94 | 0.8666 |
| Subject | -15.4 | 2.94 | < .0001 | -20.3 | 2.94 | < .0001 | -16.6 | 2.94 | < .0001 |
| Verb | -33.4 | 2.94 | < .0001 | -25.7 | 2.94 | < .0001 | -31.2 | 2.94 | < .0001 |
| Post-Verb | -17.4 | 2.94 | < .0001 | -25.5 | 2.94 | < .0001 | -27.6 | 2.94 | < .0001 |
| Object | - | - | - | - | - | - | -28.6588 | 2.94 | < .0001 |

Proportions of gaze fixations towards the visual target were calculated as a function of ROI, trial type (test, control) and verb type (unaccusative, unergative, transitive).

subject in the spoken sentence. These results also showed that control trials constituted a baseline for fixations into the visual target, which was relevant for the next step in data analysis.

Subsequent steps in data processing were carried out following Koring et al. [19]. The position of the eye fixation in the visual display was down-sampled every 20 ms. We computed the proportion of looks at the visual target in each time bin along with the presentation of the linguistic stimuli. We calculated the difference of proportion of looks to the target in the test and control conditions for each trial. This was done to obtain the proportion of looks to the visual target that reflected a reactivation effect.

## 3.2 Data analysis

Koring et al. [19] examined gaze-fixation data within two time windows: (a) the verb frame, consisting of the time period occurring from -600 ms (mean verb onset) until 1000 ms after verb offset; and (b) the post-verb frame, consisting on the time period occurring from 200 ms until 1700 ms after verb offset. The authors shifted the timeline 200 ms down throughout the entire data frame to account for the time that the human eye needs to program and initiate movement in reaction to auditory stimuli [53, 67].

In our case, we conducted an analysis of our data in three different time windows: (a) the verb frame, (b) the post-verb frame, and (c) the global post-verbal frame. First, we established the verb frame, ranging from -370 ms until 1230 ms after verb offset. The reason for this change with respect to the initial value for the verb frame in Koring et al. [19] was that the average verb length in our experiment was 370 ms. Thus, we determined the initial value for our verb frame as -370 ms to maintain comparable regions of interest (in terms of duration) with those of the verb frame in Koring et al. [19]. Second, we established a post-verb frame, which was kept identical to Koring et al. [19], ranging from 200 ms until 1700 ms after verb offset. Lastly, we established a global post-verbal frame, not included in Koring et al. [19], ranging from 200 ms until 3968 ms after verb offset. This third time frame went from verb offset to the mean sentence offset in unaccusative and unergative sentences. This time window was included to conduct a post-hoc analysis of the gaze-fixation patterns in a broader time frame. It should be noted that sentences with unaccusative and unergative verbs had a mean duration of 3768 ms after verb offset, whereas sentences with transitive verbs had a mean duration of 4830 ms after verb offset (due to the additional sentential object). However, data analysis using the Growth Curve Analysis technique (henceforth GCA) requires all conditions to have comparable durations in order to fit unbiased orthogonal shapes. Because we analyzed gaze-fixation data from 200 until 3968 ms after verb offset across all conditions, the global post-verbal analysis did not include the processing of the second argument in transitive sentences (i.e., the sentential object).

We created three separate models using the GCA technique [68] following Koring et al. [19]. The first model was created to analyze the verb frame; the second, to examine the post-

verb frame; and the third, to explore the global post-verbal frame. The first two models were identical in structure to those reported in Koring et al. [19]. The third model was identical to the one used to analyze the post-verb frame in Koring et al. [19]. The dependent variable was the difference of proportion of looks to the target in the test condition minus the proportion of looks to the target in the control condition for each pair of test and control trials. The independent variables were sentence condition (unaccusative, unergative, transitive) and orthogonal polynomials. The first model contained linear and quadratic polynomials, whereas the second and third models included linear, quadratic, cubic and quartic polynomials. Random effects by subject and random slopes by subject per each condition were also included in the three models. The analysis was conducted using R Statistic 3.6.2 software [64] using the lme4 package [65].

## 4. Results

### 4.1 Verb frame

The time course of the difference on the proportion of looks between target and control trials for each condition was modelled using the terms intercept, linear and quadratic. The models of the three conditions were compared, taking transitive verbs as the baseline. The goodness of fit of the models was analyzed using Akaike's Information Criterion (AIC) and the change in the -2 times log-likelihood was used to assess the significance of the additional terms in the nested models. Results are presented in Table 3. There was a significant effect of verb type on the intercept, but not on the linear and quadratic terms. This means that the difference between verb types was based on the average height of the curve but not on the progression of the curve. In Fig 2, we present the fit of the data in the most complex model (i.e., the quadratic model).

Pairwise comparisons between the three conditions are shown in Table 4. The pattern of fixations towards the visual target was significantly lower in the unaccusative condition in comparison with the transitive and unergative conditions. By contrast, the average height of the curve did not differ between the transitive and unergative conditions. Pairwise comparisons across sentence conditions revealed no difference between the conditions in the linear term; there was an effect on the quadratic component limited to the comparison between unaccusative and unergative subjects. There was a significant interaction between unergative subjects and the quadratic orthogonal. Per condition analysis showed that unergative subjects had a significant positive quadratic component ($ß = 0.273$; $t = 2.62$; $p = .012$), signaling meaningful fall and rise in the proportion of looks towards the visual target. Critically, unaccusative

**Table 3. Analysis of the goodness of fit of the model in the verb frame.**

| Model fit | | | | |
|---|---|---|---|---|
| **Model** | **AIC** | **-2LL** | **Chisq** | **$p <$** |
| **Base** | -9976.3 | 5004.20 | - | - |
| **Condition** | - | - | - | - |
| **x Intercept** | -9982.9 | 5009.40 | 10.5573 | .005 ** |
| **x Linear** | -9981.9 | 5011.00 | 3.0246 | .220 |
| **x Quadratic** | -9982.1 | 5013.00 | 4.1764 | .123 |

Analysis of the goodness of fit of the curves in the three conditions (unaccusative, unergative, transitive) across the verb frame (from -370 ms before verb offset until 1230 ms after verb offset). Transitive verbs are taken as a baseline. Asterisks signal levels of significance: $p < .05$ (*), $p < .01$ (**), and $p < .001$ (***).

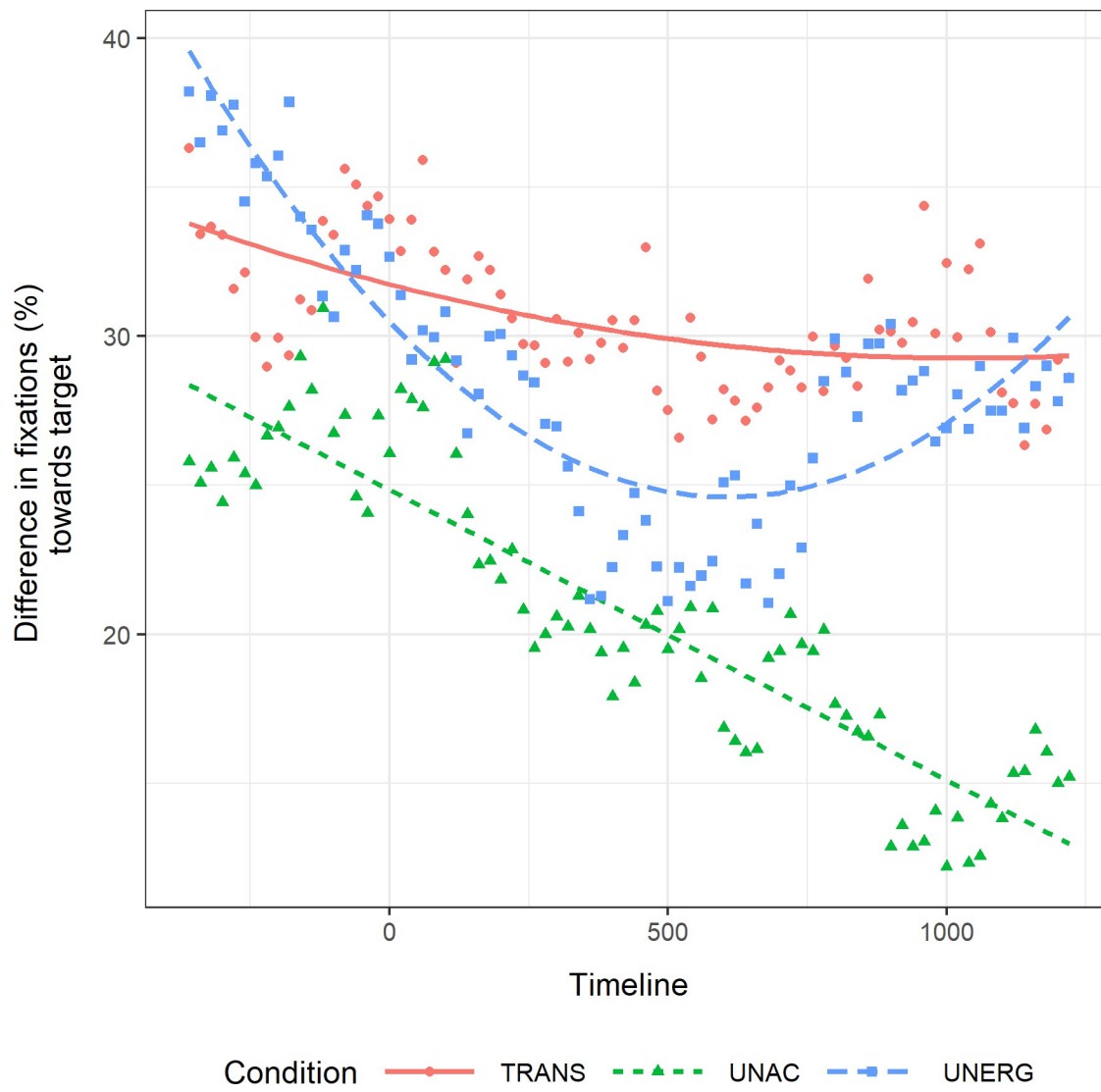

**Fig 2. Difference of fixations to visual target in the verb frame.** Difference of mean percentage of fixations to the visual target between test and control trials across the verb frame (from -370 ms before verb offset until 1230 ms after verb offset). The timeline is centered in 0 ms, which corresponds to verb offset.

**Table 4. Pairwise comparisons of the models of the curves in the verb frame.**

| Parameter estimates | | | | | | | | | |
|---|---|---|---|---|---|---|---|---|---|
| | Transitive—Unaccusative | | | Transitive—Unergative | | | Unergative—Unaccusative | | |
| Model | Est. | t | p < | Est. | t | p < | Est. | t | p < |
| Condition | - | - | - | - | - | - | - | - | - |
| x Intercept | -0.099 | -2.975 | .00473** | -0.020 | -0.620 | .539 | -0.078 | -2.595 | .01278* |
| x Linear | -0.286 | -1.592 | .114 | -0.117 | -0.68 | .498 | -0.168 | -0.894 | .373 |
| x Quadratic | -0.039 | -0.294 | .770 | 0.233 | 1.58 | .12 | -0.272 | -2.021 | .046* |

Pairwise comparisons of the models of the curves across the three conditions (unaccusative, unergative, transitive) across the verb frame.

subjects had a significant negative linear component ($\beta$ = -0.402; $t$ = -2.943; $p$ = .00517) by themselves. Transitive subjects did not show a significant linear ($\beta$ = -0.115; $t$ = -0.967; $p$ = .339) nor quadratic ($\beta$ = 0.039; $t$ = 0.312; $p$ = .757) component by themselves.

## 4.2 Post-verb frame

For the post-verb frame, we modelled the data as in the verb frame while also including a cubic and quartic term, following Koring et al. [19]. Transitive subjects were taken as a baseline to compare the models in the three conditions. Fig 3 shows the model fit of the experimental data. Again, the goodness of fit of the models was analyzed using Akaike's Information Criterion (AIC) and the change in the -2 times log-likelihood was used to assess the significance of the additional terms in the nested models (see Table 5). There was an effect of condition only on the intercept. That is, the average height of the curve was different across verb conditions,

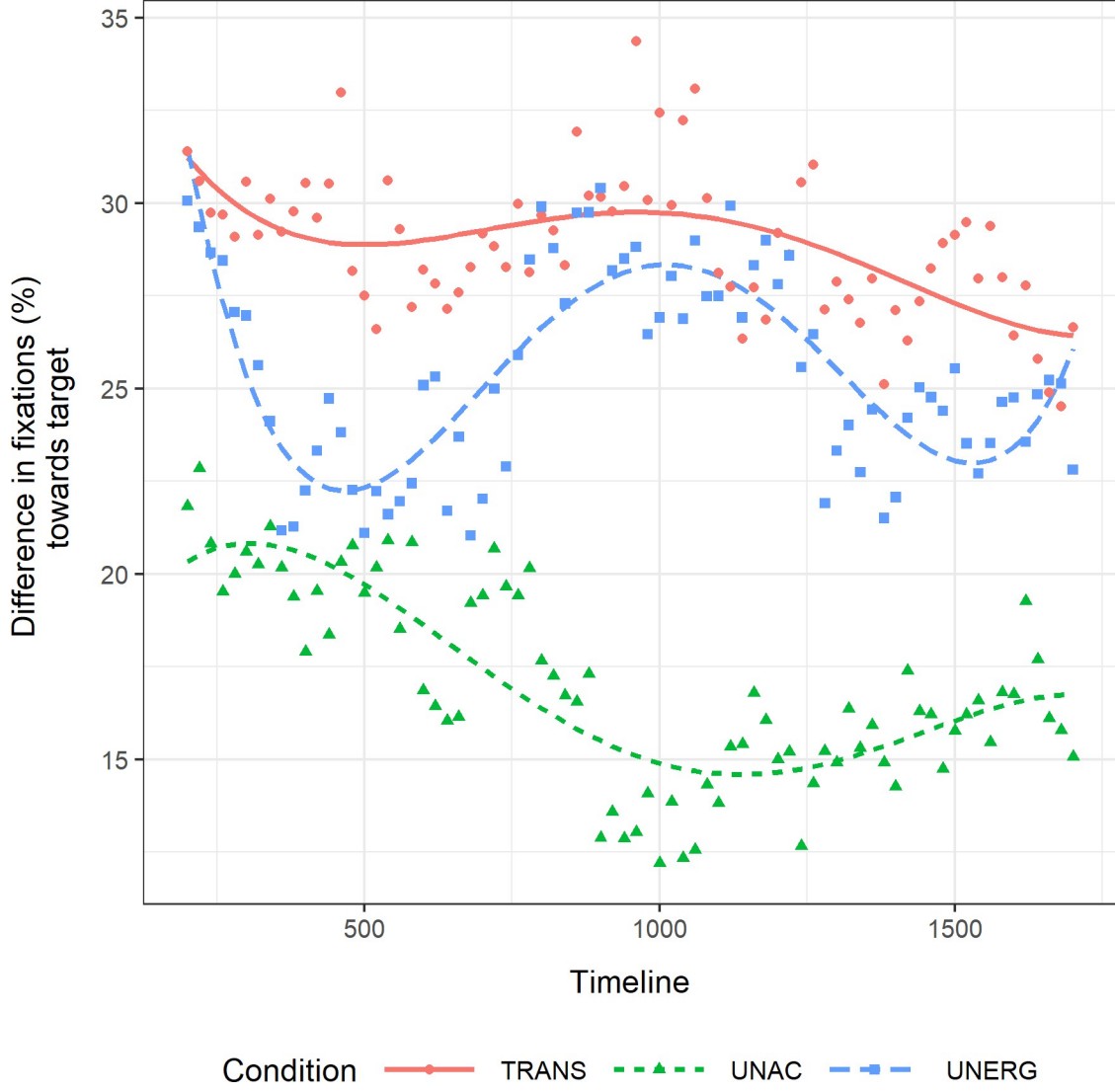

**Fig 3. Difference of fixations to visual target in the post-verb frame.** Difference of mean percentage of fixations to the visual target between test and control trials across the post-verb frame (from 200 ms until 1700 ms after verb offset). The timeline is centered in 0 ms, which corresponds to verb offset.

**Table 5. Analysis of the goodness of fit of the model in the post-verb frame.**

| Model fit | | | | |
|---|---|---|---|---|
| **Model** | AIC | -2LL | Chisq | $p <$ |
| **Base** | -13257 | 6664.6 | - | - |
| **Condition** | - | - | - | - |
| **x Intercept** | -13267 | 6671.5 | 13.764 | .001026* |
| **x Linear** | -13263 | 6671.6 | 0.224 | .894028 |
| **x Quadratic** | -13260 | 6672 | 0.823 | .662344 |
| **x Cubic** | -13258 | 6672.8 | 1.5985 | .449674 |
| **x Quartic** | -13259 | 6675.6 | 5.579 | .061435 |

Analysis of the goodness of fit of the curves in the three conditions (unaccusative, unergative, transitive) across the post-verb frame (from 200 ms until 1700 ms after verb offset). Transitive subjects are taken as a baseline.

but there was no interaction between any verb condition and a particular orthogonal component.

The results of the pairwise comparisons across conditions are shown in Table 6. Unaccusative, unergative and transitive subjects did not differ from each other in the linear term. All three conditions had a negative slope. However, the parameter estimates for the different conditions showed that none of the falls was significant by itself (transitive subjects: $\beta$ = -0.071; $t$ = -0.422; $p$ = .675; unergative subjects: $\beta$ = -0.012; $t$ = -0.1; $p$ = .9209; unaccusative subjects: $\beta$ = -0.145; $t$ = -1.342; $p$ = .186).

There was no effect on the quadratic term. Transitive and unergative conditions had a negative estimate (i.e., a rise followed by a fall), whereas the unaccusative condition had a positive quadratic estimate (i.e., a fall followed by a rise). However, none of the parameter estimates for the different conditions was significant by itself (transitive subjects: $\beta$ = -0.0386; $t$ = -0.519; $p$ = .606; unergative subjects: $\beta$ = -0.0574; $t$ = -0.634; $p$ = .5295; unaccusative subjects: $\beta$ = 0.10701; $t$ = 1.129; $p$ = .265) and verb conditions did not differ in this component, as shown in Table 6.

There was no effect on the cubic term. Transitive and unergative conditions had a negative cubic estimate, whereas the unaccusative condition had a positive estimate. Still, the cubic component was not significant by itself across conditions (transitive subjects: $\beta$ = -0.03511; $t$ = -0.385; $p$ = .702; unergative subjects: $\beta$ = -0.087; $t$ = -1.201; $p$ = .2362; unaccusative subjects: $\beta$ = 0.037; $t$ = 0.469; $p$ = .641), and verb conditions did not differ from each other in this term.

There was an effect on the quartic term. Unergative subjects had a positive quartic component ($\beta$ = 0.158; $t$ = 2.454; $p$ = .0181), unlike transitive ($\beta$ = 0.0301; $t$ = 0.539; $p$ = .593) and

**Table 6. Pairwise comparisons of the models of the curves in the verb frame.**

| Parameter estimates | | | | | | | | | |
|---|---|---|---|---|---|---|---|---|---|
| | **Transitive—Unaccusative** | | | **Transitive—Unergative** | | | **Unergative—Unaccusative** | | |
| **Model** | Est. | t | $p <$ | Est. | t | $p <$ | Est. | t | $p <$ |
| **Condition** | | | | | | | | | |
| **x Intercept** | -0.118 | -3.821 | .000381*** | -0.034 | -0.985 | .329 | -0.083 | -2.82 | .007** |
| **x Linear** | 0.073 | -0.417 | .678 | 0.059 | 0.379 | .707 | -0.133 | -0.834 | .4086 |
| **x Quadratic** | 0.145 | 1.241 | .218 | -0.018 | -0.164 | .87 | 0.164 | 1.3 | .1974 |
| **x Cubic** | 0.072 | 0.648 | .520 | -0.052 | -0.481 | .634 | 0.125 | 1.296 | .2017 |
| **x Quartic** | -0.066 | -0.856 | .394 | 0.128 | 1.53 | .13 | -0.195 | -2.389 | .0196* |

Pairwise comparisons of the models of the curves across the different conditions (unaccusative, unergative, transitive) across the post-verb frame.

unaccusative ($\beta$ = -0.03679; $t$ = -0.628; $p$ = .533) subjects. Unergative and unaccusative subjects were the only ones differing from each other in this term. Unergative subjects showed a clear three bend shape, i.e., a fall followed by a rise followed by a fall followed by a late rise.

### 4.3 Global post-verbal frame

Overall, our results showed different gaze-fixation patterns associated with processing different verb types. Thus, we replicated the general finding in Koring et al. [19], since we also found a processing difference in subject reactivation between unaccusative and unergative predicates. We report finding evidence of subject reactivation at verb position in the three experimental conditions, as indicated by gaze-fixation differences between test and control trials. Still, the magnitude and trajectory of said reactivation were different across unergative, transitive and unaccusative conditions. Unlike Koring et al. [19], we failed to find any peak (late or early) in the reactivation of the unaccusative subject. Overall, Koring et al. [19] reported a negative quadratic effect for the unergative condition in the verb frame, signaling an early peak of reactivation around 300 ms after verb offset, as well as a significant positive quartic effect for the unaccusative condition in the post-verb frame, signaling a late peak of reactivation around 950 ms after verb offset. By contrast, our results showed a positive quadratic effect for the unergative condition and a negative linear effect for the unaccusative condition in the verb frame, as well as a positive quartic effect for the unergative condition in the post-verb frame. Hence, we found a peak of reactivation of unergative subjects around 1000 ms after verb offset, but no reactivation peak for unaccusative subjects.

It could be the case that the pattern of increased and decreased fixations towards the visual target, which signals the reactivation and deactivation of the argument, was affected by experimental confounds. One such confound could be the speed of auditory presentation of the linguistic stimuli. Although our stimuli were identical to that of Koring et al. [19] in terms of syllabic length, the duration of the stimuli in milliseconds diverged between studies. Koring et al. [19] did not facilitate average speech rate in their study, but their linguistic stimuli lasted on average more than 4000 ms from sentential subject NP onset until verb offset. In our study, the equivalent linguistic stimuli lasted around 2000 ms. Hence, we concluded that our linguistic stimuli were presented at a faster speech rate than in Koring et al. [19], although our materials were recorded at a comfortable speaking rate. All other factors being equal, faster presentation of the preverbal linguistic stimuli could have affected argument reactivation in two ways. First, it could be the case that there was not enough time for listeners to deactivate the preverbal argument in their mental representation before the verbal information was presented. If that were the case, it could be possible that our measurements captured a continuous activation of the argument, rather than a reactivation, after verb offset. This would not account for differences in the pattern of subject reactivation between conditions, since it would affect all conditions equally. Second, faster speech rate could cause a delay in argument reactivation to a later time point than in Koring et al. [19] due to deferral in language processing [69]. Once again, this would not account for differences between conditions, since it would affect all conditions equally. To rule out the possibility that the peak in subject reactivation for the unaccusative condition was observed at a later point in our data, we conducted a post-hoc analysis including a wider time window, the global post-verbal frame, ranging from 200 ms to 3968 ms after verb offset (see section 3.2. for further details).

Transitive subjects were taken as a baseline to compare the models in the three conditions. The results of the model comparison are reported in Table 7. Adding new terms did not improve the explicative level of the model as measured by -2 times log-likelihood statistics and Akaike's Information Criterion (AIC). The model with the simplest linear interaction fitted

**Table 7. Analysis of the goodness of fit of the model in the global post-verbal frame.**

| Model fit | | | | |
|---|---|---|---|---|
| Model | AIC | -2LL | Chisq | $p<$ |
| Base | -22200 | 11136 | - | - |
| Condition | - | - | - | - |
| x Intercept | -22210 | 11143 | 13.4607 | .001194 |
| x Linear | -22210 | 11145 | 4.6201 | .099256 |
| x Quadratic | -22212 | 11148 | 5.3715 | .068172 |
| x Cubic | -22208 | 11148 | 0.0401 | .980147 |
| x Quartic | -22205 | 11148 | 1.0300 | .597493 |

Analysis of the goodness of fit of the curves in the three conditions (unaccusative, unergative, transitive) across the global post-verbal frame (from 200 ms until 3968 ms after verb offset). Transitive subjects are taken as a baseline.

the data better than the most complex quartic interaction. However, we report the quadratic model results, following Koring et al. [19]. Fig 4 shows the model fit of the experimental data.

Pairwise comparisons across conditions based on the full quadratic model are presented in Table 8. The magnitude of the fixations towards the visual target was greater in both transitive and unergative conditions than in the unaccusative condition. There was no difference between transitive and unergative conditions. Crucially, there was a significant interaction with the quadratic component in the comparison between unergative and unaccusative conditions. Per condition analyses aligned with these results. Unergative subjects had a negative quadratic component, which was significant by itself ($ß = -0.392$; $t = -2.764$; $p = .0083$), whereas linear ($ß = -0.296$; $t = -1.81$; $p = .077$), cubic ($ß = -0.160$; $t = -1.156$; $p = .254$) and quartic ($ß = 0.088$; $t = 0.881$; $p = .383$) components were not significant by themselves. In the transitive and unaccusative conditions, none of the falls nor the rises were significant by themselves, as we can see for the transitive linear component ($ß = -0.100$; $t = -0.507$; $p = .615$), the transitive quadratic component ($ß = -0.009$; $t = -0.049$; $p = .961$), the transitive cubic component ($ß = -0.107$; $t = -0.775$; $p = .443$), the transitive quartic component ($ß = -0.050$; $t = -0.541$; $p = .591$), the unaccusative linear component ($ß = 0.014$; $t = 0.074$; $p = .942$), the unaccusative quadratic component ($ß = 0.077$; $t = 0.557$; $p = .58$), the unaccusative cubic component ($ß = -0.144$; $t = -1.276$; $p = .209$), and the unaccusative quartic component ($ß = 0.067$; $t = 0.538$; $p = .593$).

## 4.4 Resampling analysis

In previous work also closely replicating Koring et al. [19], Huang and Snedeker [46] claimed that the GCA is inadequate to study argument reactivation within the VWP. Based on their resampling analysis results, these authors stated that growth curve models are highly anti-conservative and report many false positives, even more than expected due to chance. Considering the results reported by Huang and Snedeker [46], and in order to check the validity of our analysis, we conducted a resampling analysis on the same data frame that we used for data modelling. We randomly switched the condition label of each subject's trials, always preserving the random assignation of the condition to the complete trial of each individual so that time-series dependencies would endure. Afterwards, we conducted the GCA described in section 3.2 on the resampled data frame and registered the results. This procedure was repeated 1000 times. We applied the resampling analysis in all three time windows: the verb frame, the post-verb frame and the global post-verbal frame (i.e., the time window selected for the post-hoc analysis). Results are shown in Table 9.

Regarding the criteria for interpretation, we assume a 5% chance of false-positive result when alpha is set at 0.05. Attending to this, if our model were valid (i.e., conservative), the

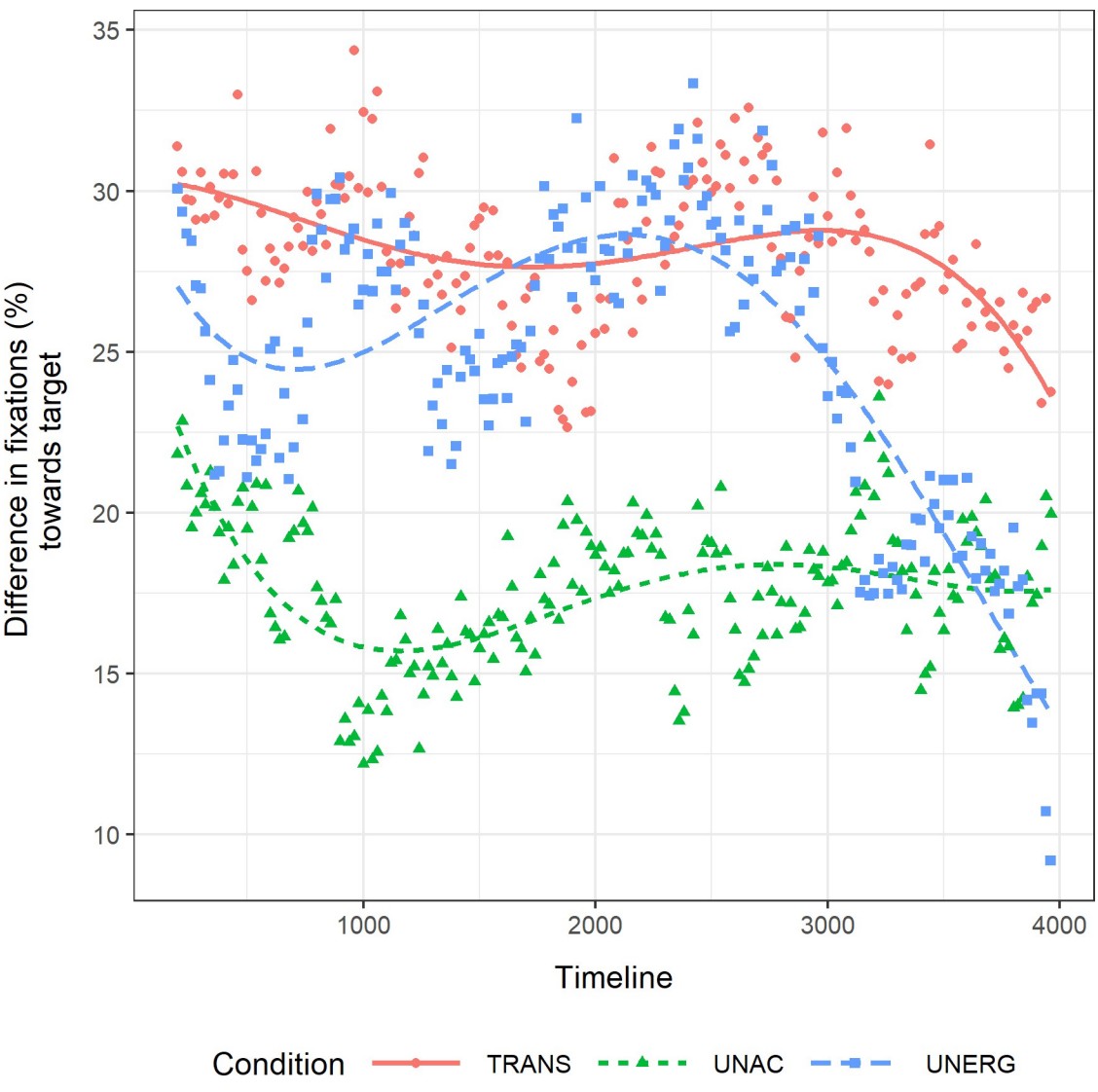

**Fig 4. Difference of fixations to visual target in the global post-verbal frame.** Difference of mean percentage of fixations to the visual target between test and control trials across the global post-verbal frame (200 ms until 3968 ms after verb offset).

matrix should not produce significant results above 5% of the time. As shown in Table 9, the overall probability of obtaining a significant effect was found in 3.80% to 7.30% of the resampled data. This means that our models were slightly anti-conservative, but still adequate for analysis. Following Huang and Snedeker [46], we also ordered the *p*-values obtained in our sample of 1000 reshufflings, selecting the 50th smallest value. If our model adequately captured the probability of false positive under the null hypothesis, this value should be around 0.05. As shown in Table 9, our data adequately captures this probability, as the provided *p*-values are around the estimated 0.05.

## 5. Discussion

In this study we investigated argument structure processing by exploring how thematic role (agent or theme) affects subject reactivation. We conducted a close replica of Koring et al. [19]

**Table 8. Pairwise comparisons of the models of the curves in the global post-verbal frame.**

| Parameter estimates | | | | | | | | | |
|---|---|---|---|---|---|---|---|---|---|
| | Transitive—Unaccusative | | | Transitive—Unergative | | | Unergative—Unaccusative | | |
| Model | Est. | t | $p <$ | Est. | t | $p <$ | Est. | t | $p <$ |
| Condition | - | - | - | - | - | - | - | - | - |
| x Intercept | -0.105 | -4.59 | 3.64e-05*** | -0.033 | -1.214 | .2312 | -0.071 | -3.056 | .0038** |
| x Linear | 0.115 | 0.524 | .603 | -0.195 | -0.864 | .3907 | 0.311 | 1.408 | .1648 |
| x Quadratic | 0.086 | 0.384 | .702 | -0.383 | -1.923 | .0598 | 0.470 | 2.454 | .0164* |
| x Cubic | -0.036 | -0.224 | .824 | -0.052 | -0.312 | .7562 | 0.016 | 0.097 | .9234 |
| x Quartic | 0.118 | 0.762 | .448 | 0.139 | 1.072 | .2872 | -0.020 | -0.136 | .8926 |

Pairwise comparisons of the models of the curves across the different conditions (unaccusative, unergative, transitive) across the global post-verbal frame.

(and partially of Huang and Snedeker [46]), investigating the reactivation of subjects after verb presentation in unaccusative, unergative and transitive sentences in Spanish. We measured the time course and magnitude of subject reactivation in three different time windows: (a) the verb frame, which was centered around the presentation of the verb; (b) the post-verb frame, which was centered around the presentation of the post-verbal Adjunct; and (c) the global post-verbal frame, which comprised a broader time frame going from verb offset until around 4000 ms after verb offset.

We entertained two hypotheses related to the time course and the magnitude of subject reactivation. On the one hand, the UH [1] predicts a late peak in the reactivation of the unaccusative subject compared to unergative and transitive subjects, in line with previous eye-tracking research [19]. The late reactivation effect for unaccusative subjects is expected to occur around 950 ms after verb offset according to various previous findings [7, 8, 10, 19, 67]. Koring et al. [19] argue that the UH makes another prediction for eye-tracking data in the VWP: a larger reactivation of unaccusative subjects than unergatives, due to their greater processing cost of the former. Instead, we argue that the magnitude of argument reactivation is an indication of attention or preference, not of processing cost. On the other hand, we considered the AFH [20–22]. The AFH predicts a larger reactivation of agent subjects (unergative and

**Table 9. Resampling analysis across the three time windows.**

| | Verb frame | | Post-verb frame | | Global post-verb frame | |
|---|---|---|---|---|---|---|
| Term | False positives | p-value for true alpha level of 0.05 | False positives | p-value for true alpha level of 0.05 | False positives | p-value for true alpha level of 0.05 |
| UNERG | 5.60% | 0.0448 | 5.20% | 0.0329 | 5.90% | 0.0419 |
| UNAC | 5.20% | 0.0457 | 5.10% | 0.0489 | 6.20% | 0.0408 |
| Linear*UNERG | 6.90%* | 0.034 | 5.20% | 0.0456 | 4.50% | 0.0530 |
| Linear*UNAC | 6.50%* | 0.0351 | 6.90%* | 0.0479 | 6.80%* | 0.0394 |
| Quadratic*UNERG | 4.30% | 0.0502 | 7.10%* | 0.0358 | 4.60% | 0.0536 |
| Quadratic*UNAC | 4.90% | 0.0570 | 6.90%* | 0.0351 | 3.80% | 0.0650 |
| Cubic*UNERG | - | - | 4.20% | 0.0613 | 5.10% | 0.0477 |
| Cubic*UNAC | - | - | 5.30% | 0.0435 | 6.70%* | 0.0382 |
| Quartic*UNERG | - | - | 7.20%* | 0.0279 | 5.40% | 0.0452 |
| Quartic*UNAC | - | - | 7.30%* | 0.0303 | 7.00%* | 0.0374 |

False positives and p-values for true alpha level of 0.05 were calculated from a sample of 1000 reshufflings. The instances in which the false positive rates were significantly greater than expected by Fisher's exact test are marked with an asterisk.

transitive conditions) than theme subjects (unaccusative condition), due to a greater saliency of or preference for agents compared to themes.

To test our hypotheses, we monitored the gaze-fixation patterns of 44 Spanish native speakers while they were simultaneously presented with an auditory sentence and a visual display with four images; one of them was semantically related to the subject in test trials, but not in control trials. We found that specific linguistic stimuli (e.g., *mouse*) drove gaze fixations to semantically-related visual objects (e.g., *cheese*), in line with previous VWP studies [51, 52]. Crucially, not only did hearing *mouse* trigger gaze fixations towards a depiction of a cheese, but the auditory presentation of the verb (e.g., *fall*) also triggered gaze fixations towards the related visual object. This finding indicates that the reactivation of previously presented arguments follows the auditory presentation of the verb in order to integrate the verb and its arguments in a single mental representation, as found also in Koring et al. [19].

At the verb frame, we found several differences across verb conditions in both the magnitude and time course of subject reactivation. The verb frame was the earliest time frame in our analysis, as it was centered on the presentation of the verb. Regarding magnitude, agent subjects (unergative and transitive conditions) reactivated to a greater degree than theme subjects (unaccusative condition). That is, the overall height of the curve that indicates looks into the visual target was greater for agent subjects than for theme subjects throughout the entire verb frame. Koring et al. [19] also found that the magnitude of reactivation was greater for unergative subjects than for unaccusatives in this time frame. Regarding time course, unergative subjects displayed a reactivation effect peaking around 1000 ms after verb offset, whereas unaccusative subjects showed a decay of activation throughout the entire verb frame. The trajectory of the curve displayed a fall of activation followed by a rise in the case of unergative subjects, but only a fall in the case of unaccusative subjects. This converges with Koring et al. [19], since they also found only unergative subjects exhibiting an early peak in reactivation in the verb frame. We found no differences when comparing the unergative and transitive conditions in this time frame. Taken by themselves, the results in this time frame can be accounted for by both the AFH and the UH. For this reason, it is necessary to combine the results from the verb frame with the ones from the post-verb frame in order to adjudicate between the two hypotheses.

At the post-verb frame, we again found differences across conditions in both the magnitude and time course of subject reactivation. The post-verb frame was centered on the presentation of the post-verbal adjunct immediately following the verb. Agent subjects (unergative and transitive conditions) also showed a greater reactivation than theme subjects (unaccusative condition): the overall height of the curve was larger for agents than for themes throughout the entire post-verb frame. Regarding time course, unergative subjects displayed two peaks of reactivation: one around 1000 ms after verb offset, and another one around 1700 ms after verb offset. Note that we cannot state that the maximum peak of reactivation was located at 1700 ms after verb offset, since the analyzed time frame ends at 1700 ms. Still, a significant reactivation effect for the unergative condition was captured starting from around 1500 ms until the end of the post-verb frame. That means that the trajectory of the curve of unergative subjects displayed a rise after verb offset, a subsequent fall and a subsequent rise toward the end of this time frame. These rises indicate a peak in the reactivation of unergative subjects. We found no differences when comparing the unergative and transitive conditions in this time frame. Unlike Koring et al. [19], we did not find a late peak in the reactivation of unaccusative subjects at the post-verb frame. During this time frame, the unaccusative condition showed a relatively low and constant pattern of post-verbal activation, without any rises in the trajectory of the curve to signal a peak in reactivation after the verb. Hence, we cannot state that the unaccusative pattern of gaze fixations constitutes a late reactivation effect.

At the global post-verbal time frame, the post-hoc analysis showed that agent subjects (unergative and transitive conditions) displayed a greater magnitude of reactivation than theme subjects (unaccusative condition). Once again, we found no evidence of a late peak in the reactivation of the unaccusative subject during this time frame. This indicates that the absence of a late peak in the reactivation of unaccusative subjects in our results was not due to a faster speech rate of the spoken sentences in our experiment.

The combined results in the verb, post-verb and global post-verbal time frames support the predictions of the AFH, but not those of the UH. We found a greater magnitude of reactivation in agent subjects than in themes, a result we interpret as evidence that listeners devoted more attentional resources towards agents than themes. However, contrary to UH predictions and Koring et al. [19], we did not observe a late peak in the reactivation of unaccusative subjects in the post-verb frame. Unaccusative subjects did not undergo a late reactivation after the presentation of the verb, but rather showed a decay in activation at the verb frame and then continued to be slightly activated in a relatively low and constant manner throughout the post-verb time frame. Lastly, we found no difference in the processing of unergative and transitive subjects in either time frame. This further indicates that the pattern of subject reactivation is guided by the thematic role of the subject.

In line with this interpretation, we also suggest that previous findings adjudicated to the UH may also be accounted for by the AFH. For example, the late reactivation of the unaccusative subject reported by Koring et al. [19], which they interpret as evidence of object-to-subject syntactic movement, was still lower in magnitude than the one observed for the unergative condition in their study [19]. Moreover, Koring et al. [19] also found a difference in the magnitude of reactivation across conditions in all time frames, which can be observed in the significant difference in intercept of the curves. That is, their overall finding that agent subjects reactivate to a greater degree than theme subjects can be accounted for by the AFH alone, without appeal to the UH.

Overall, we did not replicate the findings reported in Koring et al. [19] regarding the time course of subject reactivation in the unaccusative and unergative conditions. Whereas Koring et al. [19] found an early peak in the reactivation of unergative subjects and a late peak in the reactivation of unaccusative subjects, we found an early peak in the reactivation of unergative and transitive subjects (agents) after the verb, yet failed to find a late peak in the reactivation of unaccusative subjects (themes). This aspect of our results coincides with the findings reported in Huang and Snedeker [46], also a close replica of Koring et al.'s [19] study, who also failed to replicate the patterns of subject reactivation reported in Koring et al. [19]. Given the concerns about the method's validity expressed in Huang and Snedeker [46], we performed a resampling analysis of our data. Results validated our methods and the methods employed by Koring et al. [19] for testing processing differences of subject reactivation in different argument structures. We thus conclude that GCA of VWP data constitutes a valid method to investigate the processing of argument structure.

Our results provide new evidence concerning the processing of argument structure and thematic roles in sentence comprehension, since we found different reactivation patterns depending on the thematic role of the subject (agent or theme). Our main finding is the difference in the magnitude of subject reactivation across conditions. Subjects with an agent role (unergative and transitive conditions) displayed a post-verbal reactivation of greater magnitude than subjects with a theme role (unaccusative condition) in all time frames. These results are fully compatible with the predictions generated by the AFH, which claims that the first ambiguous NP in a sentence will preferably be interpreted as an agent. This hypothesis can also account for the differences between unaccusative and unergative predicates without necessarily involving distinct syntactic structures, as the UH crucially claims. Our results are not compatible

with the UH, since we failed to observe a late peak in the reactivation of the unaccusative subject after the verb. Because fixation data in the VWP reveals cognitive attention and activation of elements within the participant's mental state, we take our results to constitute solid evidence of a greater attentional preference towards agents, shown by the larger reactivation effect we report.

## 6. Conclusion

In this study we report new evidence of processing differences between agent and theme subjects in Spanish by means of eye tracking in the VWP. We presented participants with SV(O) spoken sentences with transitive, unergative and unaccusative verbs, while monitoring the time course of gaze fixations into the visual display. Subjects of transitive and unergative verbs are agents while subjects of unaccusative verbs are themes. In test trials, the visual target (e.g., *cheese*) was semantically related to the sentential subject (e.g., *mouse*). We measured the time course and magnitude of the activation of the sentential subject across the presentation of the spoken sentence. We found that agent subjects (transitive and unergative conditions) underwent a post-verbal reactivation of greater magnitude than theme subjects (unaccusative condition).

Our main finding lies in a difference in the magnitude of the reactivation effect, not in the time course of the effect. We found differences across verb conditions in all time frames. Unergative and transitive subjects, both agents, displayed a larger reactivation effect than unaccusative subjects (themes). We did not find any significant difference in subject reactivation between the unergative and the transitive conditions, suggesting that agent subjects share a common pattern of reactivation regardless of transitivity. We interpret this difference in the magnitude of reactivation as an indication of the amount of attentional resources or cognitive preference directed towards sentential subjects during sentence comprehension. In other words, as soon as thematic roles could be determined (i.e., when participants heard the verb), we observed an attentional preference towards agents over themes.

Our findings thus fully meet the predictions made by the AFH. We found an early reactivation of the unergative subject after the verb, but crucially, not a late one for the unaccusative subject. This is contrary to Koring et al. [19], who reported a late reactivation of unaccusative subjects at the post-verb frame, which they interpret as evidence for syntactic movement as predicted by the UH. Our findings can be accounted for by the AFH without resorting to the UH, and we argue that previous experimental findings interpreted as evidence for the UH can also be accounted for by the AFH, following an attentional preference for agents compared to themes rooted in human cognition.

## Supporting information

**S1 Table. Mean ratings of strongly-related nouns.** Online norming study results showing the mean ratings for strongly related noun-noun pairs.
(DOCX)

**S2 Table. Mean ratings of weakly-related nouns.** Online norming study results showing the mean ratings for weakly-related noun-noun pairs.
(DOCX)

**S3 Table. Mean ratings of weakly-related verb-noun pairs.** Online norming study results showing the mean ratings for weakly-related verb-noun pairs.
(DOCX)

## Author Contributions

**Conceptualization:** Beatriz Gómez-Vidal, Itziar Laka.

**Data curation:** Miren Arantzeta.

**Formal analysis:** Miren Arantzeta, Jon Paul Laka.

**Investigation:** Beatriz Gómez-Vidal.

**Methodology:** Beatriz Gómez-Vidal, Miren Arantzeta.

**Resources:** Itziar Laka.

**Supervision:** Itziar Laka.

**Validation:** Jon Paul Laka.

**Writing – original draft:** Beatriz Gómez-Vidal, Miren Arantzeta.

**Writing – review & editing:** Beatriz Gómez-Vidal, Miren Arantzeta, Itziar Laka.

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
