## [Decision Letter · Decision Letter 0]

11 Apr 2022

PONE-D-21-39055Subjects are not all alike: Eye-tracking the Agent Preference in SpanishPLOS ONE

Dear Dr. Gómez-Vidal,

Thank you for submitting your manuscript to PLOS ONE. After careful consideration, we feel that it has merit but does not fully meet PLOS ONE’s publication criteria as it currently stands. Therefore, we invite you to submit a revised version of the manuscript that addresses the points raised during the review process.

We look forward to receiving your revised manuscript.

Kind regards,

Andriy Myachykov, PhD

Academic Editor

PLOS ONE

Journal Requirements:

Reviewers' comments:

Reviewer's Responses to Questions

**Comments to the Author**

1. Is the manuscript technically sound, and do the data support the conclusions?

Reviewer #1: Yes

Reviewer #2: Yes

2. Has the statistical analysis been performed appropriately and rigorously? 

Reviewer #1: Yes

Reviewer #2: Yes

3. Have the authors made all data underlying the findings in their manuscript fully available?

Reviewer #1: Yes

Reviewer #2: No

4. Is the manuscript presented in an intelligible fashion and written in standard English?

Reviewer #1: Yes

Reviewer #2: Yes

5. Review Comments to the Author

Reviewer #1: PONE-D-21-39055

Summary: The paper presents the results of a single visual world study that looked at the reactivation of subjects following a verb. Results showed that unaccusatives had less reactivation compared to unergatives and transistives. These condition differences were interpreted via AFH. In contrast, the unexpected lack of peaks in the eye movement were interpreted as not supporting the UH.

Two caveats: (a) I was not previously familiar with the Koring or H&S papers (at least not in much detail) and (b) I’m also not intimately familiar with GCA procedures, and so I cannot really comment on the re-sampling section of the results.

Evaluation: I found the paper to be generally well written and clear. My biggest comment about the structure would be that the discussion seems a bit repetitive and could be cut by at 1 page. I did not see any methodological flaws or problems with the interpretation of the data. In my opinion, there is only one major issue that needs to be addressed and it is speech rate. I picked up on this point before it was mentioned in the results as a potential problem. I recommend revise and re-submit.

Major Points:

1. Methods: If you went to the work to calculate the number of syllables of all items and length in msec then why not report it for the different conditions. And also, it is good practice to include the number of syllables per second for an accurate reporting of speech rate (see Fernandez et al. QJEP 2020). Speech rate is surprisingly consistent (within speaker) and so I wouldn’t expect condition differences to be a confound in your study. However, knowing the speech rate can be a key factor in the resolution of competing/mixed findings (e.g. koring vs. snedeker).

2. Results and Discussion: I agree with the speed of presentation issue, see comment above. However, I disagree with the conclusions on page 29. That is, that the peak should simply be later for a faster speech rate. In reality the effect of fast speech is a washing out of the linking between speech processing and direction of visual attention. BTW, the same also occurs for very slow speech. Again, I would highly recommend a detailed consultation of the Fernandez QJEP paper. The bottom line is that I think you need to (1) clearly report what your speech rate was, and (2) be very very clear in the discussion that speech rate should be factored in/considered in the interpretation. Is it possible to follow up on the H&S materials, were they fast or slow?

3. There seems to be very little information about the fillers. I’d assume that some actually named objects shown in the array?

Minor Points:

1. The word derivation is used at several places in the introduction, but that word seems a bit misleading. Personally, I would tend to use derivation to refer to production. Because this study is comprehension, then shouldn’t the expected longer processing be because of the added interpretive steps with identifying the gap and assigning an argument to that gap.

2. Line 94: provide reference for reactivation of argument at trace.

3. Line 201: define test vs. control

4. Line 216: report the comparisons of syllables in ROIs

5. Figure 4 should be more similar to others (axes, legend, etc.)

Reviewer #2: In this paper, the authors reported an eye-tracking study that assessed the hypotheses of processing unaccusative and unergative predicates. On the one hand, the unaccusative hypothesis argues that the processing of unaccusatives requires an extra step and thus takes longer to re-access the subject of the sentence relative to unergatives. On the other hand, the agent first hypothesis champions an across-the-board processing advantage for agents. The two hypotheses make different predictions on the re-activation of the subject argument after accessing the verb. Concretely, the unaccusative hypothesis predicts an early reactivation for unergative subjects, and a late reactivation for unaccusative subjects, whereas the agent first hypothesis predicts a stronger reactivation effect on unergative subjects. The author tested these predictions in an eye-tracking experiment wherein native Spanish speakers read Spanish unaccusative or unergative sentences while viewing a visual display with four items. In the visual display, one item is semantically related to the subject of the critical items. The fixation on the semantically related item was measured. The authors found that unergatives showed a stronger reactivation effect while no reactivation effect could be found for unaccusatives. The findings support the predictions from the agent first hypothesis.

Overall, I think this is a very solid paper that tried to test the validity of an established study. The author did a nice job motivating the research question and had very clear predictions regarding the results of the experiment. The paper is well-structured and the logic is consistent throughout. I recommend accepting this paper pending minor revisions.

I don't have any major concerns about the paper. The only major-ish comment is about the organization of the Introduction. If I understand correctly, most parts of the Introduction centered around the unaccusative hypothesis, whereas the agent first hypothesis was rarely discussed (the term was not even mentioned). Indeed, the second paragraph talked about the agent/theme distinction, but it seems a bit out of place and broke the flow of the writing. So I would suggest having a part that specifically introduces the theoretical motivation of the agent first hypothesis in the Introduction, such that the discussion of the two hypotheses can be more balanced.

Minor comments

P7 Line 156. what is an "NVN" strategy? Please explain in the text.

P9 Line 208. A weird spacing.

P10 Participants and procedure. I wonder how the listening-while-viewing task really worked. First of all, is there any cover story for the task? Did the participant have to do anything other than listen to the sentence and view the picture? Second, there was no fixation before the onset of the ROIs, so we have no idea what the status of eye-movement was at the onset of the ROIs. Would you please tell me whether the issues above might potentially influence the results of the experiment?

P25 Line 574-576. "In our study, we do not consider this prediction as one of the possible readings of the UH for eye-tracking data in the VWP". Are you referring to the results of your own experiment?

References. I think you need to provide the DOIs of the references.

6. PLOS authors have the option to publish the peer review history of their article (what does this mean?). If published, this will include your full peer review and any attached files.

Reviewer #1: No

Reviewer #2: No

---

## [Author Response · Author response to Decision Letter 0]

2 Jun 2022

We would like to thank the academic Editor and Reviewers for their enriching and thorough commentary of our manuscript. We have responded to all reviewer and editor comments in their Decision Letter in a separate file, which we have submitted to this platform. Please find a detailed response to each point raised by the Reviewers in our file titled "Response to Reviewers".

---

## [Decision Letter · Decision Letter 1]

15 Jul 2022

Subjects are not all alike: Eye-tracking the Agent Preference in Spanish

PONE-D-21-39055R1

Dear Dr. Gómez-Vidal,

We’re pleased to inform you that your manuscript has been judged scientifically suitable for publication and will be formally accepted for publication once it meets all outstanding technical requirements.

Kind regards,

Andriy Myachykov, PhD

Academic Editor

PLOS ONE

Additional Editor Comments (optional):

Reviewers' comments:

Reviewer's Responses to Questions

**Comments to the Author**

1. If the authors have adequately addressed your comments raised in a previous round of review and you feel that this manuscript is now acceptable for publication, you may indicate that here to bypass the “Comments to the Author” section, enter your conflict of interest statement in the “Confidential to Editor” section, and submit your "Accept" recommendation.

Reviewer #1: All comments have been addressed

Reviewer #2: All comments have been addressed

2. Is the manuscript technically sound, and do the data support the conclusions?

Reviewer #1: Yes

Reviewer #2: Yes

3. Has the statistical analysis been performed appropriately and rigorously? 

Reviewer #1: Yes

Reviewer #2: Yes

4. Have the authors made all data underlying the findings in their manuscript fully available?

Reviewer #1: Yes

Reviewer #2: Yes

5. Is the manuscript presented in an intelligible fashion and written in standard English?

Reviewer #1: Yes

Reviewer #2: Yes

6. Review Comments to the Author

Reviewer #1: I only had one major issue, and it concerned speech rate. The authors seem to have gone over the top in addressing this concern and incorporating it into the paper as a potential issue within the results but also across studies.

Reviewer #2: In an eye-tracking study, the authors tested two hypotheses of processing unaccusative and unergative predicates: the unaccusative hypothesis (UH) and the agent first hypothesis (AFH). The former predicts distinctive timings of reactivation (unergative subjects precede unaccusative subjects), while the latter assumes that the magnitudes of the reactivation effects are different (unergative subjects show a stronger effect). The results of a Visual World experiment demonstrated that, when comprehending Spanish SV(O) sentences while viewing visual displays, native Spanish speakers showed a systematic variation of the fixation on the targets that is more compatible with the prediction of AFH.

The authors did a great job in improving the manuscript. All of my comments have been addressed. I believe this paper is ready for publication.

The only minor suggestion is about the notion of the Visual World Paradigm. This term was introduced on page 4. But an explanation of VWP didn't appear until the next page. My concern is that those who have no idea what this paradigm is might not know how eye fixations in VWP index sentence processing (in particular reactiation). It could be nice if you can put the basic assumption of VWP immediately after the introduction of this paradigm.

Minor comments

1. Line 618 "as found also in Koring et al." -> as was also found

7. PLOS authors have the option to publish the peer review history of their article (what does this mean?). If published, this will include your full peer review and any attached files.

Reviewer #1: No

Reviewer #2: No

---

## [Editor Report · Acceptance letter]

26 Jul 2022

PONE-D-21-39055R1 

Subjects are not all alike: Eye-tracking the Agent Preference in Spanish 

Dear Dr. Gómez-Vidal:

I'm pleased to inform you that your manuscript has been deemed suitable for publication in PLOS ONE. Congratulations! Your manuscript is now with our production department. 

Kind regards, 

on behalf of

Dr. Andriy Myachykov 

Academic Editor

PLOS ONE